# ResSwinUnet3D: Developing a New Residual-Based SwinUnet3D Model for Enhanced 3D Medical Image Segmentation

B. Vaibhav Mallya[1], Micky C. Nnamdi[1], J. Ben Tamo[1], Yishan Zhong[1], Wenqi Shi[2], May D. Wang[1]

[1] *Georgia Institute of Technology*, Atlanta, USA
[2] *University of Texas Southwestern Medical Center*, Dallas, USA
{bmallya3, mnnamdi3, jtamo3, yzhong307, maywang}@gatech.edu, wenqi.shi@utsouthwestern.edu

*Abstract*—Accurate segmentation of 3D medical images remains a significant challenge due to complex anatomical variations, low contrast between adjacent structures, and the computational burden associated with volumetric data. Conventional deep learning models often encounter vanishing gradients and limited feature propagation in deep architectures, particularly when handling large-scale 3D volumes. To address these issues, this paper presents ResSwinUnet3D, a residual SwinUnet3D architecture for 3D medical image segmentation that combines vision transformers, convolutional neural networks, and residual connections. The proposed model extends the SwinUnet3D design by introducing residual blocks between the encoder and decoder components to mitigate the vanishing gradient problem and improve information flow through deep layers. Experiments were conducted on three datasets: BraTS 2020, BraTS 2021, and Synapse Multi-Organ CT Segmentation. On the BraTS 2020 dataset, our model achieved Dice Similarity Coefficients of 0.9170, 0.8539, and 0.8030 for whole Tumor, Tumor Core, and Enhancing Tumor regions, respectively. For the BraTS 2021 dataset, our model achieved Dice scores of 0.9211, 0.9200, and 0.8924 for Whole Tumor, Tumor Core, and Enhanced Tumor, respectively. On the Synapse Multi-Organ CT Segmentation dataset, ResSwinUnet3D attained a mean Dice score of 0.8276 across 13 organ classes. With the integration of residual blocks, our model achieves a 5–20% overall improvement in performance compared to SwinUNet3D and other similar models such as Attention UNet and UNETR across the previously specified datasets and evaluation metrics. Gradient-weighted Class Activation Mapping analyses further showed that residual connections produce interpretable activation maps, clarifying the model's decision process. These findings suggest that ResSwinUnet3D offers a robust and efficient solution for volumetric segmentation across diverse organs and imaging modalities.

*Index Terms*—3D medical image segmentation, class activation maps, decoder, encoder, residual blocks, vanishing gradients, vision transformers

## I. INTRODUCTION

Accurately outlining structures in 3D data, known as three-dimensional (3D) volumetric image segmentation, plays a pivotal role in medical imaging and many other fields [1]. This process plays a vital role in applications like treatment planning, disease diagnosis, and quantitative analysis [2], [3]. Despite its importance, this task remains difficult because of the complexity of 3D spatial information, the need to maintain consistency across multiple slices, and the significant computational demands posed by large-scale volumetric datasets [4], [5]. Recent progress in deep learning has greatly influenced medical image segmentation. By modeling contextual information and intricate spatial dependencies, modern learning approaches enable segmentation that is both more precise and more efficient [6].

Convolutional Neural Networks (CNNs) have traditionally served as the foundation of image processing tasks [7]–[9], particularly in image segmentation [10], [11]. Models such as 3D U-Net [12], [13] and V-Net [14] have achieved success in processing entire 3D volumes while effectively preserving spatial context across all dimensions [10]. By leveraging hierarchical structures to capture local and global features, these networks achieve robust segmentation, with 3D U-Net setting a benchmark in medical imaging through its effective spatial dependency modeling [15]. Similarly, V-Net's efficiency and accuracy have made it a preferred choice for many applications. However, CNN-based methods are constrained by their localized receptive fields, which limit their ability to model long-range dependencies. To address these challenges, researchers have drawn inspiration from the Natural Language Processing (NLP) domain, where Transformer models have demonstrated remarkable effectiveness in capturing long-range dependencies [16], [17]. Adapting Transformer architectures for computer vision tasks has led to significant advancements, as seen in models like the Vision Transformer (ViT) [18] and the Shifted windows (Swin) Transformer [19]. These models utilize self-attention mechanisms to learn contextual relationships across entire input sequences, a characteristic that has proven highly beneficial for image analysis. Building on these innovations, researchers have introduced hybrid architectures that address the shortcomings of CNNs and pure Transformer models. UNEt TRansformers (UNETR) [20], for instance, integrates Transformers as the encoder within a "U-shaped" network design, effectively capturing long-range context information. UNETR++ [21] enhances this approach by introducing a paired attention block to efficiently learn spatial and channel features simultaneously. While CNNs excel in capturing local features, Transformer-based models are superior in modeling global dependencies, prompting the development of hybrid solutions such as SwinUNETR [22] and SwinUnet3D [23]. These architectures combine the strengths of both paradigms, enabling the learning of detailed and global features while addressing the shortcomings of their predecessors.

In this work, we introduce ResSwinUnet3D, an enhanced architecture derived from SwinUnet3D [23]. While keeping the encoder and decoder intact from the original model, residual blocks are added to address the problem of vanishing gra-

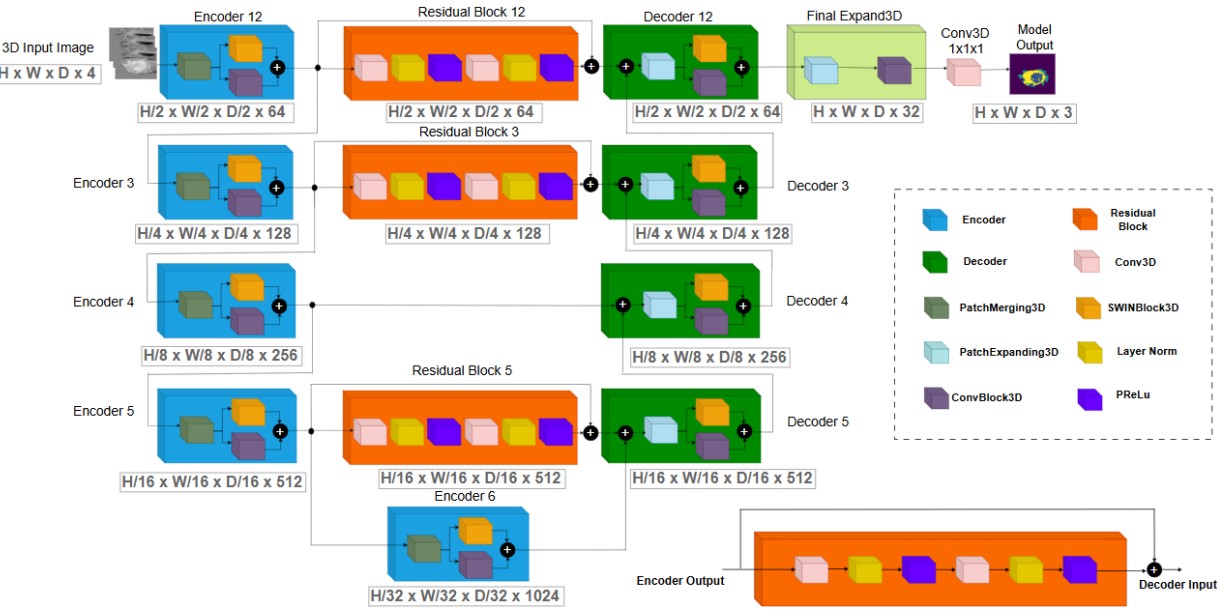

Fig. 1. Illustrates the ResSwinUnet3D architecture, a 3D U-shaped network that combines convolutional layers and Swin Transformer. Its encoder–decoder structure incorporates skip connections to effectively preserve spatial details. The bottom section highlights the architecture of the residual block, featuring ConvBlock3D, Layer Norm, and PReLU activation for efficient feature propagation. The section on the right provides a color key for the various blocks used in the architecture.

dients, enabling better learning and information propagation through deep layers. To demonstrate how these enhancements influence the model, we employ Gradient-weighted Class Activation Mapping (Grad-CAM) [24] to create interpretable heatmaps that visualize the decision-making process of the model. The primary contributions of our work are twofold:

- **Architecture Enhancement**: Extended the state-of-the-art SwinUnet3D by incorporating residual blocks, addressing vanishing gradient issues, and boosting the stability and performance of volumetric image segmentation.
- **Model Interpretability**: Utilized Grad-CAM to generate interpretable heatmaps, bridging the gap between traditional performance metrics and offering deeper insights into the model's decision-making process.

## II. METHODOLOGY

### A. Network Architecture

ResSwinUnet3D architecture comprises an encoder, a residual block, a jump connection, and a decoder, designed to enhance the network's efficiency and robustness in processing 3D image data. The architecture's design features a residual block strategically positioned between the encoder and decoder. This innovation mitigates the vanishing gradient issue frequently encountered in deep learning, enabling improved information propagation through the network's deeper layers. As illustrated in Figure 1, the residual block diagram is centrally located, with a block color key provided at the bottom for clarity. The encoder and decoder designs are directly inspired by the architecture proposed in [23], ensuring a strong foundation for effective feature extraction and reconstruction.

The encoder in ResSwinUnet3D consists of the Patch-Merging3D blocks, 3D Convolution (ConvBlock3D) units, and SWINBlock3D units. The primary functionality of the

PatchMerging3D block is to downsample the input image while increasing the number of channels, effectively preparing the data for subsequent processing. The ConvBlock3D units are designed to learn local dependencies within the image, while the SWINBlock3D units capture global dependencies through advanced self-attention mechanisms. Together, these components ensure that the encoder efficiently extracts and processes both local and global features. In contrast, the PatchExpanding3D block in the decoder upsamples the image by increasing its spatial dimensions and decreasing the number of channels, thereby reversing the downsampling effect of the PatchMerging3D block. The ConvBlock3D units in the decoder focus on learning refined local features, while the SWINBlock3D units capture global dependencies to ensure the preservation of context and structure.

Assume an input 3D image $X$ of dimension $H \times W \times D$, PatchMerging3D operates by dividing the 3D input image into non-overlapping voxel blocks of size $P \times P \times P$. Each voxel block is flattened into a one-dimensional vector of size $V = P^3$, which is then linearly transformed to a vector of length $N$, effectively encoding the input into multiple $\frac{H}{P} \times \frac{W}{P} \times \frac{D}{P}$ tokens. Each token $T$ has a length of $N$. In each stage $i$ of the encoder, the PatchMerging3D block downscales the tokens $T^i$ by a factor of 2 in each dimension using a sequence of Conv3D and LayerNorm (LN) operations, producing the output $A^i$. $A^i$ is subsequently fed into the Shifted window Transformer block (SWINBlock3D) and ConvBlock3D unit simultaneously. In the ConvBlock3D unit, $A^i$ is processed through a series of Parameterized ReLU (PReLU), LN, and $1 \times 1 \times 1$ Conv3D layers, repeated twice, to extract local feature representations, yielding $C_e^i$. In contrast, the SWINBlock3D unit comprises four main components. First, an LN block and a window multi-head self-attention 3D (WMSA3D) module compute attention

scores between tokens within non-overlapping sub-windows. The computation of $X_{t_1}^i$ uses the self-attention mechanism in the WMSA3D block and is described as follows:

$$\tilde{A}^i = LN(A^i) \quad (1)$$

$$Q_{\tilde{A}^i} = W^Q \cdot \tilde{A}^i, \quad K_{\tilde{A}^i} = W^K \cdot \tilde{A}^i, \quad V_{\tilde{A}^i} = W^V \cdot \tilde{A}^i \quad (2)$$

$$atten_{\tilde{A}^i} = \text{Softmax}\left(\frac{Q_{\tilde{A}^i} \cdot K_{\tilde{A}^i}^T}{\sqrt{d_k}}\right) \cdot V_{\tilde{A}^i} \quad (3)$$

$$X_{t_1}^i = \tilde{A}^i + atten_{\tilde{A}^i} \quad (4)$$

Here, $W^Q$, $W^K$, and $W^V$ are learnable query, key, and value weight matrices, respectively. The term $d_k$ represents the dimensionality of the matrix $K_{\tilde{A}^i}$, while $atten_{\tilde{A}^i}$ is a matrix containing the self-attention scores. This mechanism enables the WMSA3D block to capture contextual information by modeling relationships between tokens within each window. Secondly, the features produced in the first step are refined to compute $X_t^i$ using an LN block and a Multi-Layer Perceptron (MLP). The third unit of the SWINBlock3D includes an LN block and a Shifted-Window Multi-Head Self-Attention 3D (SWMSA3D) module. While the WMSA3D module is effective for calculating self-attention scores within individual windows, it cannot compute self-attention scores between tokens located in adjacent windows. To address this limitation, the SWMSA3D module is employed. In the SWMSA3D module, the input tokens are shifted by $s$ units along all three spatial dimensions. By default, $s$ is set to half the window size. This shifting mechanism enables the computation of self-attention scores across adjacent windows. However, it also introduces two potential issues: (1) inconsistencies in window sizes and (2) an increased number of windows. To mitigate these problems, the attention scores for tokens that were originally in non-adjacent windows before cyclic shifting are filtered out. As a result, the SWMSA3D module retains only the attention scores of tokens that were initially located in adjacent windows before the shift. This ensures consistency and computational efficiency. The output $X_{t_2}^i$ is computed within the 3SWMSA3D module. The input $X_{t_1}^i$ is first normalized, then transformed into query, key, and value matrices. Scaled dot-product attention is applied, and the resulting attention output is added back to the normalized input to yield $X_{t_2}^i$. This allows the model to capture dependencies across adjacent windows while maintaining stability through residual connections.

$$X_{t_2}^i = \tilde{X_{t_1}^i} + atten_{X_{t_1}^{\tilde{i}}} \quad (5)$$

Finally, another LN block and an MLP refine the output, yielding the features $S_e^i$. The final step of the encoder involves the concatenation of the two outputs, $C_e^i$ and $S_e^i$, which complement each other to form $E^i$. This combined representation encapsulates both local and global feature information, ensuring a comprehensive encoding of the input data. The output of the encoder, $E^i$ (Encoders 1-5), is passed through the residual block, which consists of a jump connection and a stacked combination of PReLU, LN, and a $1 \times 1 \times 1$ Conv3D block, repeated twice, to produce $R^i$ (Res Blocks 12, 3 and 5). Starting at the Encoder 5 stage, its output $E^5$ is concatenated with $R^5$ to form $I^5$. $I^5$ is then passed through the PatchExpanding3D block to upsample its resolution by a factor of 2, resulting in $U^5$. The upsampled feature $U^5$ is fed into the SWINBlock3D and a ConvBlock3D unit, undergoing similar transformations as $A^i$ in Equations 1 through 5, to produce $S_d^5$ and $C_d^5$, respectively. These features are concatenated to produce $D^5$ (Decoder 5). Similarly, for the remaining Decoder stages j, $D^j$ is concatenated with the corresponding output from the residual block or encoder stage, progressing until $D^1$ is produced. Finally, $D^1$ is passed through a FinalExpand3D block (includes PatchExpanding3D block + PReLU block) and $1 \times 1 \times 1$ Conv3D block to generate the model output $Y$.

The number of multi-head self-attention mechanisms in the encoder stages 1, 2, 3, 4, and 5 are 3, 6, 9, 12, and 15, respectively. Additionally, the decoder stages 4, 3, 2, and 1 use 12, 9, 6, and 3 multi-head self-attention mechanisms, respectively. Each stage of the encoder and decoder blocks employs 2 SWINBlock3D units.

### B. Loss Functions

In this study, we adopt a hybrid Dice Cross Entropy loss, which integrates the complementary benefits of the Dice loss and the Cross Entropy loss. This combined objective balances region-level overlap with voxel-level classification accuracy, making it particularly effective for segmentation tasks involving imbalanced class distributions.

$$\mathcal{L}_{Dice+CE}(\mathbf{X}, \mathbf{Y}) = 1 - \sum_{i=1}^{C}\left(\frac{2\sum_{j=1}^{M} X_{j,i} \cdot Y_{j,i}}{\sum_{j=1}^{M} X_{j,i}^2 + \sum_{j=1}^{M} Y_{j,i}^2} + \sum_{j=1}^{M} X_{j,i}\log Y_{j,i}\right), \quad (6)$$

where $C$ represents the total number of classes, $M$ is the number of voxels, and $X_{j,i}$ and $Y_{j,i}$ represent the probabilities of the ground truth and the predicted output at voxel $j$ and class $i$, respectively.

### C. Gradient-Weighted Class Activation Mapping

Gradient-weighted Class Activation Mapping (Grad-CAM) [24] is a widely used explainable AI technique that provides visual insights into how deep models arrive at their predictions. In our study, Grad-CAM is employed to justify the integration of residual blocks within the proposed architecture. Specifically, the technique highlights the most influential spatial regions in the input that guide the network's output, allowing us to visualize the decision pathway. These heatmaps reveal how residual connections improve gradient flow and strengthen information transmission across layers, ultimately helping to address vanishing gradient issues while supporting more effective feature representation.

### III. EXPERIMENTAL RESULTS AND DISCUSSION

#### A. Datasets

In this study, three datasets were utilized: BraTS 2020 [26]–[28], BraTS 2021 [26], [27], [29] and Synapse Multi-Organ CT Segmentation datasets [30] for experimental evaluations.

1) **The BraTS 2020 and BraTS 2021** contain multi-institutional Magnetic resonance imaging (MRI) scans

TABLE I

PERFORMANCE ANALYSIS OF THE PROPOSED MODEL FOR BRATS 2020 AND BRATS 2021 DATASETS; $DSC-$ DICE SIMILARITY COEFFICIENT ; $S_e-$ SENSITIVITY ; $S_p-$ SPECIFICITY ; $A_c-$ ACCURACY ; $IoU-$ INTERSECTION OVER UNION; $P_r-$ PRECISION $WT-$ WHOLE TUMOR ; $TC-$ TUMOR CORE ; $ET-$ ENHANCED TUMOR

| Metric | BraTS 2020 | | | BraTS 2021 | | |
|---|---|---|---|---|---|---|
| | WT | TC | ET | WT | TC | ET |
| $DSC$ | $0.9170 \pm 0.0867$ | $0.8539 \pm 0.1579$ | $0.8030 \pm 0.1947$ | $0.9211 \pm 0.0887$ | $0.9200 \pm 0.1474$ | $0.8924 \pm 0.1703$ |
| $S_e$ | $0.9170 \pm 0.0599$ | $0.8592 \pm 0.0624$ | $0.7957 \pm 0.0736$ | $0.9155 \pm 0.0885$ | $0.9183 \pm 0.0646$ | $0.8850 \pm 0.0326$ |
| $S_p$ | $0.9991 \pm 0.0389$ | $0.9994 \pm 0.0462$ | $0.9997 \pm 0.0069$ | $0.9994 \pm 0.0161$ | $0.9997 \pm 0.0216$ | $0.9998 \pm 0.0534$ |
| $A_c$ | $0.9983 \pm 0.0387$ | $0.9986 \pm 0.0461$ | $0.9994 \pm 0.0069$ | $0.9987 \pm 0.0164$ | $0.9994 \pm 0.0217$ | $0.9995 \pm 0.0533$ |
| $IoU$ | $0.8507 \pm 0.0996$ | $0.7688 \pm 0.1703$ | $0.7022 \pm 0.1951$ | $0.8636 \pm 0.0957$ | $0.8691 \pm 0.1654$ | $0.8190 \pm 0.1806$ |
| $P_r$ | $0.9245 \pm 0.1196$ | $0.8831 \pm 0.2182$ | $0.7649 \pm 0.2353$ | $0.9395 \pm 0.0913$ | $0.9291 \pm 0.1866$ | $0.8759 \pm 0.2062$ |

TABLE II

COMPARISON OF DSC SCORES OF OUR MODEL WITH STATE-OF-THE-ART MODELS FOR BRATS 2020 AND BRATS 2021 DATASETS.

| Method | BraTS 2020 | | | BraTS 2021 | | |
|---|---|---|---|---|---|---|
| | WT | TC | ET | WT | TC | ET |
| UNETR [20] | $0.8999 \pm 0.1649$ | $0.8122 \pm 0.1995$ | $0.7738 \pm 0.2762$ | $0.9058 \pm 0.0957$ | $0.8950 \pm 0.1769$ | $0.8769 \pm 0.1490$ |
| UNETR++ [21] | $0.8756 \pm 0.0830$ | $0.7540 \pm 0.0899$ | $0.7030 \pm 0.0998$ | $0.8383 \pm 0.2095$ | $0.8150 \pm 0.2098$ | $0.7019 \pm 0.3189$ |
| ATTENTION UNET [25] | $0.7260 \pm 0.1767$ | $0.5180 \pm 0.1781$ | $0.7579 \pm 0.1473$ | $0.8829 \pm 0.1956$ | $0.8176 \pm 0.1664$ | $0.8403 \pm 0.1545$ |
| SWIN UNETR [22] | $0.9136 \pm 0.1428$ | $0.8531 \pm 0.2089$ | $0.8084 \pm 0.2463$ | $0.9163 \pm 0.1555$ | $0.9186 \pm 0.1557$ | $0.8964 \pm 0.1682$ |
| SwinUnet3D [23] | $0.9106 \pm 0.0830$ | $0.8511 \pm 0.1523$ | $0.7844 \pm 0.1910$ | $0.8889 \pm 0.0847$ | $0.9024 \pm 0.1438$ | $0.8688 \pm 0.1677$ |
| ResSwinUnet3D (Ours) | $0.9170 \pm 0.0867$ | $0.8539 \pm 0.1579$ | $0.8030 \pm 0.1947$ | $0.9211 \pm 0.0887$ | $0.9200 \pm 0.1474$ | $0.8924 \pm 0.1703$ |

TABLE III

COMPARISON OF DSC SCORES OF OUR MODEL WITH STATE OF THE ART MODELS FOR SYNAPSE DATASET

| Method | DSC | IoU |
|---|---|---|
| UNETR [20] | $0.7964 \pm 0.1620$ | $0.6893 \pm 0.2078$ |
| ATTENTION UNET [25] | $0.7929 \pm 0.2061$ | $0.6971 \pm 0.2401$ |
| SWIN UNETR [22] | $0.8229 \pm 0.1572$ | $0.7248 \pm 0.1973$ |
| SwinUnet3D [23] | $0.8074 \pm 0.1828$ | $0.7092 \pm 0.2156$ |
| ResSwinUnet3D (Ours) | $0.8276 \pm 0.1285$ | $0.7257 \pm 0.1814$ |

of glioma patients, with 484 cases in the 2020 release and 1,251 in the 2021 release. Each subject is provided with four MRI modalities—native T1, post-contrast T1-weighted (T1Gd), T2-weighted, and T2-FLAIR—resampled to a uniform resolution of $240 \times 240 \times 155$ voxels at $1mm^3$ spacing. The ground-truth masks delineate three tumor subregions: the necrotic/non-enhancing tumor core (NCR, label 1), peritumoral edema (ED, label 2), and the enhancing tumor (ET, label 4). By converting the original multi-class labels into a one-hot encoded multi-label framework, we facilitated faster convergence and optimized segmentation performance. Label 2 was isolated to form the Enhanced Tumor (ET) channel. The tumor core (TC) channel was constructed by merging Labels 2 and 4. The whole tumor (WT) channel was created by merging Labels 1, 2, and 4 to capture the entire extent of the tumor. We employed a logical OR function to merge the specified labels.

2) **The Synapse dataset** consists of 30 CT scans which have variable volume sizes between $512 \times 512 \times 85$ and $512 \times 512 \times 198$. Each scan is annotated for 13 abdominal structures: spleen, right kidney, left kidney, gallbladder, esophagus, liver, stomach, aorta, inferior vena cava, portal & splenic vein, pancreas, and both adrenal glands. Labels are assigned sequentially from 1 to 13.

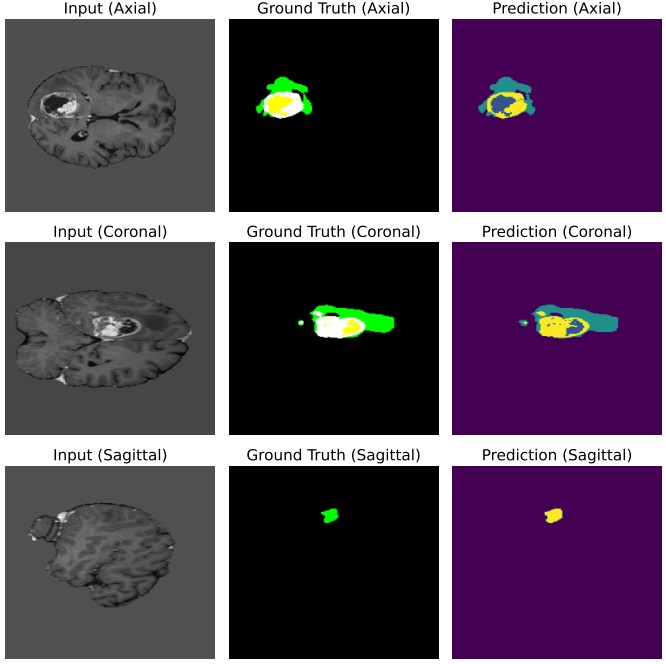

Fig. 2. BraTS dataset outputs visualized across Axial, Coronal, and Sagittal planes, comparing the input MRI images, ground truth segmentation masks, and model predictions. The results demonstrate the model's ability to accurately identify tumor regions in different anatomical views, with predictions closely matching the ground truth in all three planes.

### B. Evaluation Metrics

To assess the effectiveness of the proposed model, we employed a set of widely recognized segmentation metrics chosen to capture different aspects of performance across the datasets. These include the Dice Similarity Coefficient ($DSC$), Sensitivity ($S_e$), Specificity ($S_p$), Accuracy ($A_c$), Intersection over Union ($IoU$), and Precision ($P_r$). For the Synapse multi-

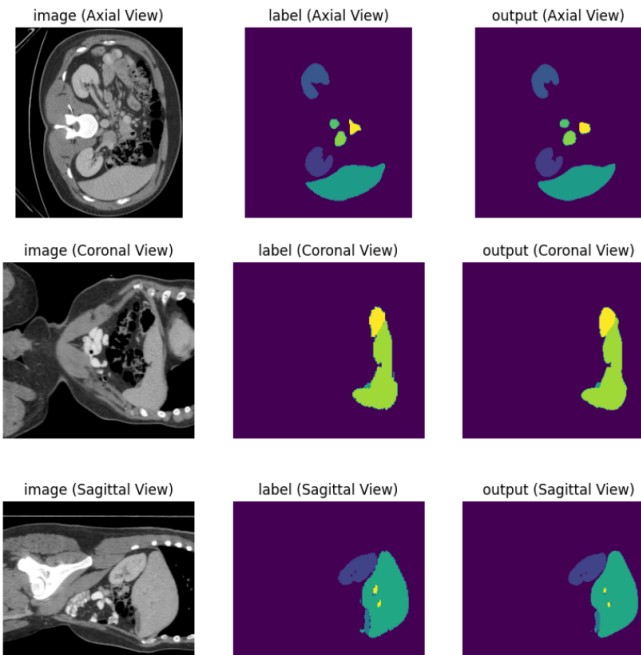

Fig. 3. Synapse dataset outputs visualized across Axial, Coronal, and Sagittal planes, showing input CT images, ground truth segmentation labels, and model predictions. The model accurately segments anatomical structures in different views, with predictions closely aligning with the labeled regions in all planes.

organ dataset, we report the average $DSC$ across all annotated organs. This aggregate score offers a concise representation of segmentation quality in multi-class settings, where consistency across multiple anatomical structures is essential.

### C. Experimental Details

We implemented our model using PyTorch [32], with preprocessing facilitated by the MONAI [33] library. The transformation between feature maps and tokens was handled using the einops [34] library. To accelerate model training, we employed the PyTorch Lightning [35] framework. To ensure uniformity and fairness, all models were trained using the same data splits, preprocessing strategies, input sizes, and loss functions. Training was conducted on an NVIDIA H100 GPU for all datasets. The learning rate was set to 0.0003, optimized using Adam with a weight decay of $1 \times 10^{-5}$, and adjusted using a cosine annealing learning rate scheduler with a maximum period of 10 epochs. Datasets were split into training, validation, and testing sets in a 60:20:20 ratio, with a fixed random seed of 42 to ensure reproducibility.

### D. Main Result

As shown in Table I and Figure 2, the model achieved a DSC of 0.9170 for WT, 0.8539 for TC, and 0.8030 for ET, demonstrating its high accuracy in segmenting the whole tumor and its subregions on BraTS 2020. $S_p$ was near-perfect across all regions, with values exceeding 0.999, indicating the model's ability to identify negative cases correctly. Similarly, the model achieved high $A_c$ across all subregions, with values nearing 1.000. $IoU$ scores further highlight robust performance, particularly for $WT$ (0.8507) and $TC$ (0.7688). On the BraTS 2021 dataset, the proposed model achieved even higher $DSC$ values

### TABLE IV
ABLATION STUDY HIGHLIGHTING THE IMPORTANCE OF THE RESIDUAL BLOCKS USING THE BRATS 2021 DATASET

| Method | DSC | IoU |
|---|---|---|
| ResSwinUnet3D + 0 Residual Block | $0.8421 \pm 0.1321$ | $0.7685 \pm 0.1438$ |
| ResSwinUnet3D + 1 Residual Block | $0.8295 \pm 0.1541$ | $0.7519 \pm 0.1448$ |
| ResSwinUnet3D + 2 Residual Block | $0.8367 \pm 0.1278$ | $0.7605 \pm 0.1380$ |
| ResSwinUnet3D + 3 Residual Block | $0.9117 \pm 0.1355$ | $0.8506 \pm 0.1472$ |

for $WT$ (0.9211) and $TC$ (0.9200), showcasing improved generalization. $S_p$ and $A_c$ remained consistent, demonstrating the model's robustness across datasets. The $IoU$ values were similarly strong, with $WT$ achieving 0.8636 and $TC$ reaching 0.8691. On the Synapse multi-organ segmentation dataset (Figure 3), the model achieved an average $DSC$ of 0.8276, highlighting its capability to generalize beyond brain tumor segmentation tasks.

### E. Comparison with State-of-the-Art Models

To further validate the performance of the proposed ResS-WINUnet3D model, we compared its results with several state-of-the-art segmentation models, including UNETR [20], Attention UNET [36], SwinUNETR [22], and SwinUnet3D [23], on the BraTS 2020, BraTS 2021, and Synapse datasets. The comparison is summarized in Tables II and III.

As shown in Table II, the proposed ResSwinUnet3D model demonstrated better performance across all segmentation tasks in both datasets. For the BraTS 2020 dataset, ResSwinUnet3D outperforms other models; however, it still faced challenges similar to other models in this $ET$ subregion, highlighting its inherent difficulty in segmentation. A consistent trend was observed in the BraTS 2021 dataset, where ResSwinUnet3D consistently delivered competitive results compared to other models. Furthermore, on the Synapse multi-organ segmentation dataset, as summarized in Table III, ResSwinUnet3D achieved a mean $DSC$ of 0.8276, outperforming UNETR, Attention UNET, SwinUNETR, and SwinUnet3D. Although SwinUNETR achieved a comparable score of 0.8229, ResSwinUnet3D demonstrated a slight improvement, effectively handling multi-organ segmentation tasks involving diverse anatomical structures.

### F. Ablation Study

Our ablation study, presented in Table IV, investigates the contribution of residual blocks in the skip connections. The results indicate that while adding one or two blocks provides limited benefit, the inclusion of all three residual blocks yields a substantial performance improvement, boosting the DSC and IoU scores by approximately 8.9% and 10.7% respectively, over the baseline without any residual blocks. This non-linear improvement suggests a synergistic effect; the complete set of residual blocks is crucial for effectively fusing the multi-scale spatial features from the encoder with the semantic features from the decoder.

### G. Explainability

As shown in Figure 4, the GradCAM visualizations reveal differences in feature representation between Encoder 12 block and Residual 12 block. The residual block, introduced as part of the proposed architecture, appears to refine the feature maps

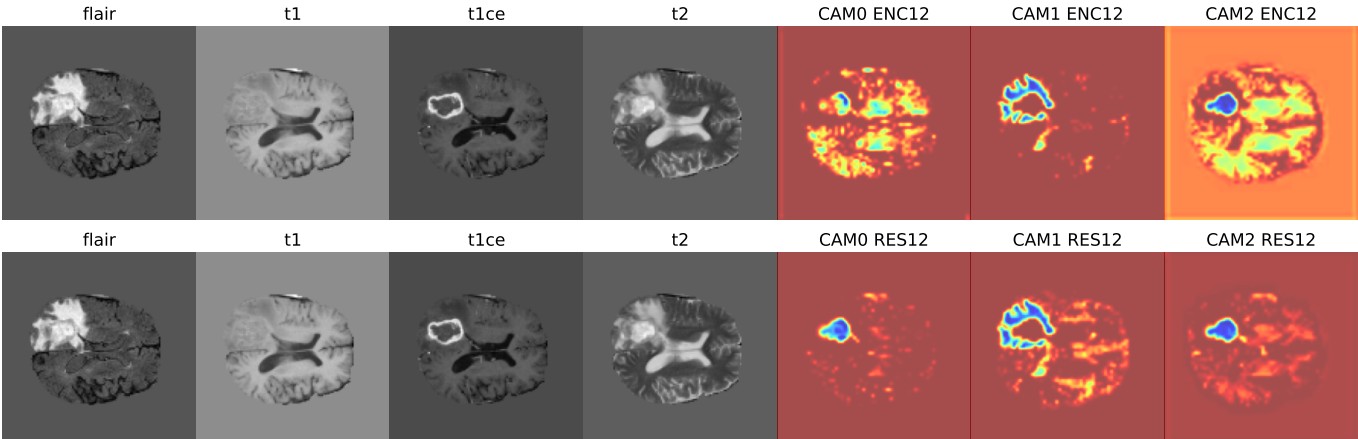

Fig. 4. GradCAM for the 3 BraTS classes. The top row shows the four modalities and GradCAM outputs for Encoder12 block; the bottom row shows the same for Residual12 block. The residual block heatmaps are more well-defined and cohesive than the encoder's, which appear more diffused. Quantitatively, the residual block achieves an average insertion Dice score area under curve (AUC) of 0.500 and a deletion Dice score AUC of 0.162 as defined by [31] (vs. insertion AUC of 0.48 and a deletion AUC of 0.163 without the residual block). Here, insertion AUC measures how quickly model confidence recovers as top-ranked regions are added back, while deletion AUC measures how rapidly confidence drops as those regions are removed, where higher insertion AUC scores and lower deletion AUC scores imply more faithfulness in the explanations.

generated by the encoder. The heatmaps for the residual block (bottom row) show more cohesive and concentrated activation patterns, particularly in regions corresponding to tumors. This suggests that the residual block may enhance the focus and precision of feature representations. On the other hand, the heatmaps for the encoder (top row) are more diffused, with less-defined activation regions, which could indicate a broader and less specific feature representation. These observations suggest that the residual block may contribute to improved spatial preservation and feature refinement, which could aid in segmentation tasks. Additionally, we perform a faithfulness analysis of the explanations for the image by calculating the Insertion/Deletion Area Under Curve (IAUC/DAUC) of the Dice score at $8 \times 8 \times 8$ voxel increments as defined in [31]. We note that GradCAM faithfulness increases due to the decrease in DAUC and the increase in IAUC when residual blocks are included. By visualizing the GradCAM heatmaps and quantifying the improvement in GradCAM's faithfulness due to the residual blocks, our analysis provides insights into how the residual block enhances feature representation and contributes to the model's overall performance.

## IV. DISCUSSION

One of the key challenges in training deep neural networks, particularly for tasks such as 3D volumetric image segmentation, is the vanishing gradient problem. This occurs when gradients become too small during backpropagation, preventing effective weight updates in deeper layers. To address this, we introduced residual blocks into the SwinUnet3D architecture. These residual blocks help maintain gradient flow, allowing for better information propagation throughout the network. The results suggest that these blocks play a significant role in improving feature representation, as evidenced by the GradCAM visualizations and quantitative metrics. The GradCAM heatmaps reveal that the residual blocks refine feature representations, producing more cohesive and concentrated activation patterns compared to the encoder

layers. This refinement is particularly beneficial for tumor boundary localization, a critical aspect of medical imaging applications. While the encoder layers display more diffused activation patterns, the residual blocks appear to focus the model's attention on relevant features, potentially improving segmentation accuracy and enhancing the interpretability of the model's predictions.

The statistical analysis further supports the robustness of the proposed model. Pearson and Spearman correlation coefficients between the predictions and ground truth labels were both 0.9179. While the integration of residual blocks successfully addresses the vanishing gradient problem and enhances feature refinement, the model's computational complexity remains a limitation. Transformer-based encoding and the processing of large input volumes require substantial computational resources, which may pose challenges in resource-constrained environments. Future work could explore optimizing the architecture to reduce computational overhead while maintaining its strong performance.

## V. CONCLUSION

In this study, we have proposed ResSwinUnet3D, a novel architecture for 3D medical image segmentation that effectively addresses the challenges of long-range dependency modeling and fine-grained detail preservation. ResSwinUnet3D integrates residual blocks into the SwinUnet3D framework, thereby combining the strengths of convolutional neural networks and transformer-based mechanisms to achieve state-of-the-art performance on the BraTS and Synapse datasets.

The addition of residual blocks plays a crucial role in enhancing feature representation, as highlighted by Grad-CAM visualizations. The heatmaps demonstrate that the residual blocks produce more cohesive and well-defined activation patterns compared to the encoder layers, suggesting improved feature localization and better preservation of spatial information. ResSwinUnet3D shows promise for applications requiring precise segmentation.

ACKNOWLEDGMENT

This research was supported in part through AI Makerspace of the College of Engineering and Partnership for an Advanced Computing Environment (PACE) at the Georgia Institute of Technology, Atlanta, Georgia, USA. It has also been supported by a Wallace H. Coulter Distinguished Faculty Fellowship, a Petit Institute Faculty Fellowship, and research funding from Amazon and Microsoft Research to Dr. May D. Wang.

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
