# OpenReview forum: "ResSwinUnet3D: Developing A New Residual-Based SwinUnet3D Model for Enhanced 3D Medical Image Segmentation"
_IEEE.org/EMBS/BHI/2025/Conference — BHI 2025_

### Official Review · Reviewer_b4pe · 2025-07-15
**Review of ResSwinUnet3D**

**Confidence:** 3
**Clarity Of Writing:** great
**Clinical Significance:** great
**Methodological Novelty:** good
**Overall Rating:** 7
**Final Rating:** 7

**Experiments And Results:**

great

**Questions For The Authors:**

In Section III A, when discussion labels and channels it is slightly unclear whether you are using these as part of the input (so they become concatenated together to form different input channels) or are you talking about these as contributing to the labels during training (are they part of the one-hot-encoded label). Can you reconsider the last few sentences in this paragraph to improve clarity?
Table II is too small, the text is not clear. Consider giving this more space and consult the style guidelines for the minimum recommended font size.
In the ablation study, it would be interesting to know in what order the residual connections were reintroduced.

**Strengths:**

The results reported in this work seem very promising and are a validation of the effectiveness of this new architecture. The robustness of the results is supported by the testing on 3 datasets (with two quite different tasks) and comparing with other published works.
The structure of the paper is logical and it contributes to the clarity of the work.

**Summary Of The Paper:**

The paper proposes an updated architecture for 3D medical image segmentation which improves performance on 3 datasets. The novel improvements include adding residual connections to a 3D U-Net architecture (that also includes SWIN transformer layers) and the application of Grad-CAM to visualize the parts of activation maps that are contributing to the model's decisions.

The addition of residual connections has a positive impact on the performance of the model on two different tasks: brain MRI scans and CT scans of 13 different organs. The paper suggests that the inclusion of these residual connections reduces the impact of vanishing gradients during training. An ablation study shows the impact of removing the residual blocks and highlights their importance in this architecture. The paper shows that Grad-CAM can be utilized to visualize the intermediate feature representations, potentially adding to the explainability of the model.

**Weaknesses:**

There are no significant weaknesses that I have identified.
Although I believe that the potential for explainability of the model has been shown in this work, more would need to be done to show the clinical significance of using Grad-CAM for explaining model decisions or potentially explaining where a model deviates from the ground truth labels.

---

### Official Review · Reviewer_5CHh · 2025-07-17
**347**

**Confidence:** 4
**Clarity Of Writing:** good
**Clinical Significance:** great
**Methodological Novelty:** good
**Overall Rating:** 6

**Experiments And Results:**

good

**Questions For The Authors:**

Please refer to the weaknesses listed above for details.

**Strengths:**

1, The integration of residual blocks into a transformer-based 3D segmentation model is well-motivated and effectively addresses vanishing gradient issues in deep networks.\
2, The model achieves competitive and often superior Dice scores across multiple datasets, including BraTS 2020/2021 and Synapse, indicating strong generalization.\
3, The ablation study directly quantifies the contribution of residual blocks, providing strong empirical justification for their inclusion.\
4, Training setup and experimental protocols are rigorously documented, with consistent preprocessing and evaluation metrics across models for fair comparison.

**Summary Of The Paper:**

This paper introduces ResSwinUnet3D, a residual-enhanced 3D hybrid architecture that combines Swin Transformers and CNNs for volumetric medical image segmentation. The key novelty lies in integrating residual blocks between encoder and decoder modules to improve gradient flow and model interpretability, showing improved performance on BraTS and Synapse datasets.

**Weaknesses:**

1, The paper lacks a clear visualization or block diagram explaining how residual blocks interface with encoder/decoder tokens at each level.\
2, Despite mentioning computational overhead, the paper does not include any quantitative comparison of training/inference time or memory usage vs. baseline SwinUnet3D.\
3, The performance gain in some cases (e.g., Synapse) is marginal; the paper would benefit from statistical significance testing to support claims of superiority.\
4, Grad-CAM AUC metrics (insertion/deletion) are mentioned without direct comparison to alternative methods or visual baselines.

---

### Official Review · Reviewer_NRo6 · 2025-07-17
**Residual Block Enhanced SwinUNet3D for Medical Image Segmentation**

**Confidence:** 5
**Clarity Of Writing:** great
**Clinical Significance:** fair
**Methodological Novelty:** poor
**Overall Rating:** 4

**Experiments And Results:**

fair

**Questions For The Authors:**

It is my belief that SotA will require one or more of:  concrete motivation, architectural ingenuity at the conceptual/mathematical level either in the representation (think voxel grids, point clouds, embeddings, state, NeRF/Gaussian Splats etc. and the tradeoffs between them) or formulation in terms of architecture like iterative refinement of superpixels(voxels), structure-aware latent space, or a different loss formulation like optimal mass transport, diffusion etc. or, distinct qualitative improvements that can be emphasized. I don't have questions in particular. What is already in the manuscript looks good to me.

Including metrics for nnUNet MedNeXT and SAM3D should placate the bulk of my concerns.

Highlighting results where the model outperforms current SotA visibly/qualitatively along with failure modes, and an explanation grounded in the math/architecture would get the paper all the way to the finish line. I give the authors my best!

**Strengths:**

- The paper is quite comprehensive and covers most bases
- Writing is clear
- Significant incremental improvement in metrics

Quite a thorough submission, there's not much to complain about what is already in the manuscript.

**Summary Of The Paper:**

The paper proposes improving the architecture of SwinUNet3D by adding residual blocks and claims to achieve SotA performance on the BraTS 2018 BraTS 2021 and Synapse benchmark datasets for 3D medical image segmentation.

**Weaknesses:**

- Limited novelty - The improvement solely from adding residual blocks verges on too-good-to-be-true considering the addition of residual blocks is relatively low hanging fruit and is all the more true in the case of the MONAI stack where it's a kwarg. This is not to detract from what the paper achieves, but to articulate the source of my skepticism. Furthermore, GradCAM is standard practice and does little to push the needle as far as novelty is concerned.
- Motivation - the vanishing gradient problem, information flow etc., as terms, get thrown around a lot in general and are not as convincing a motivation as they used to be. Ideally a slightly more principled way of organizing the story would make a world of a difference - beginning with gradient propagation analysis and visualization, establishing where the bottlenecks are and then comparing gradient flow with the incremental addition of residual blocks, plotting how performance scales with the addition of blocks etc. and then talking about how that is reflected in the quantitative/scalar metrics.
- Comparison - For a paper that claims SotA  I would expect to see comparisons with nn-Unet, MedNeXt, UNETR++, and in particular SAM3D given how much discriminative ability it exhibits for the number of parameters it is comprised of. I appreciate the effort and complexity of benchmarking models that don't have plug and play implementations like those in MONAI but without those comparisons, I would be remiss to agree with the paper's claim of SotA performance.
- Architecture diagram should include dimensions. A less saturated color palette might also help.
- Table 2 should either span both columns or be split into two tables.

---

### Official Review · Reviewer_PUFc · 2025-07-19
**Addressing vanishing gradients and limited feature propagation in deep image segmentation**

**Confidence:** 3
**Clarity Of Writing:** great
**Clinical Significance:** great
**Methodological Novelty:** great
**Overall Rating:** 7

**Experiments And Results:**

great

**Questions For The Authors:**

NA

**Strengths:**

1. The paper introduces a well-motivated extension to existing 3D medical image segmentation models by integrating residual blocks into the SwinUnet3D framework. This addition effectively addresses vanishing gradient issues, facilitating deeper and more stable network training, which is a known challenge in volumetric medical imaging.

2. The authors conduct thorough experiments on three high-quality and widely-used datasets: BraTS 2020, BraTS 2021 (brain tumors), and Synapse Multi-Organ CT (multi-organ segmentation). This provides strong evidence of the model's generalizability across both tumor and organ segmentation tasks.

3. The model achieves state-of-the-art performance, with clear and consistent improvements in Dice Similarity Coefficient (DSC) and other metrics over baselines like UNETR, Attention UNet, SwinUNETR, and the original SwinUnet3D. These improvements are particularly evident for the challenging tumor core and enhanced tumor regions.

4. The use of Grad-CAM visualization to assess and compare feature representations before and after residual blocks is a notable strength. It provides insightful evidence of improved focus and feature refinement, enhancing the paper’s transparency regarding how the model achieves its results.

**Summary Of The Paper:**

The paper proposes ResSwinUnet3D, a novel deep learning architecture for 3D medical image segmentation that integrates residual blocks into the existing SwinUnet3D framework to address vanishing gradient issues and improve feature propagation in deep neural networks. This hybrid model leverages convolutional layers and Swin Transformers to capture both local and global features efficiently. Through experiments on BraTS 2020, BraTS 2021, and Synapse Multi-Organ CT datasets, ResSwinUnet3D demonstrates superior segmentation performance compared to state-of-the-art models, achieving Dice Similarity Coefficients up to 0.9211 for whole tumor segmentation and 0.8276 for multi-organ CT tasks. The inclusion of residual connections notably enhances model interpretability, as evidenced by Grad-CAM visualizations showing clearer and more focused activation maps. An ablation study further confirms the positive impact of these residual blocks on segmentation accuracy and model robustness.

**Weaknesses:**

Only results are reported in ablation study. Further analysis is required